# Functional topography of the human entorhinal cortex

**Tobias Navarro Schröder[1]\*[†], Koen V Haak[1†], Nestor I Zaragoza Jimenez[1], Christian F Beckmann[1,2,3‡], Christian F Doeller[1]\*[‡]**

[1]Donders Institute for Brain, Cognition and Behaviour, Radboud University, Nijmegen, Netherlands; [2]Department of Cognitive Neuroscience, Radboud University Medical Centre, Nijmegen, Netherlands; [3]Oxford Centre for Functional MRI of the Brain, University of Oxford, Oxford, United Kingdom

**Abstract** Despite extensive research on the role of the rodent medial and lateral entorhinal cortex (MEC/LEC) in spatial navigation, memory and related disease, their human homologues remain elusive. Here, we combine high-field functional magnetic resonance imaging at 7 T with novel data-driven and model-based analyses to identify corresponding subregions in humans based on the well-known global connectivity fingerprints in rodents and sensitivity to spatial and non-spatial information. We provide evidence for a functional division primarily along the anteroposterior axis. Localising the human homologue of the rodent MEC and LEC has important implications for translating studies on the hippocampo-entorhinal memory system from rodents to humans.

**\*For correspondence:**
t.navarroschroeder@donders.
ru.nl (TNS); christian.doeller@
donders.ru.nl (CFD)

[†]These authors contributed equally to this work

[‡]These authors also contributed equally to this work

**Competing interests:** The authors declare that no competing interests exist.

## Introduction

The entorhinal cortex (EC)—defining the interface between the hippocampus and the neocortex (*Munoz and Insausti, 2005*)—plays a pivotal role in the integration of different sensory inputs into higher order mnemonic representations (*Eichenbaum et al., 2007*; *Moser and Moser, 2013*). In rodents—and on the basis of cytoarchitectonics—the EC is typically (*Kerr et al., 2007*; *Canto et al., 2008*; *Van Strien et al., 2009*) subdivided into two major subregions, the medial- and the lateral entorhinal cortex (MEC and LEC, respectively). The MEC receives inputs about spatial information from parahippocampal cortex (PHC) and the LEC receives item-related information from perirhinal cortex (PRC) (*Van Strien et al., 2009*; *Deshmukh and Knierim, 2011*; *Ranganath and Ritchey, 2012*; *Knierim et al., 2013*). Similar functional roles of the PHC and PRC have been described in humans (*Epstein and Kanwisher, 1998*; *Davachi et al., 2003*; *Eichenbaum et al., 2007*; *Ekstrom and Bookheimer, 2007*; *Litman et al., 2009*; *Duarte et al., 2011*; *Staresina et al., 2011*; *Martin et al., 2013*; *Vilberg and Davachi, 2013*) and relate to distinct visual processing streams (*Kravitz et al., 2011*). The differential input pattern into the rodent LEC and MEC also dovetails with a cell-type specific functional specialisation (*Eichenbaum and Lipton, 2008*). The MEC contains a high proportion of head-direction and grid cells, whose activity is modulated by running direction and spatial location, respectively (*Hafting et al., 2005*; *Sargolini et al., 2006*). In contrast, cells in the LEC respond to individual objects in the environment rather than to specific locations (*Deshmukh and Knierim, 2011*; *Tsao et al., 2013*; *Knierim et al., 2013*).

Despite a wealth of data and marked differences in structure and function of the rodent MEC and LEC evidence for their human homologue remains elusive. This hampers translational studies, which is particularly relevant in the case of Alzheimer's disease (AD) with AD pathology starting in the EC (*Braak and Braak, 1992*). Within the EC, the vulnerability to AD-related pathology is not homogeneously distributed and differs between medial and lateral strips in humans, which has been related to similar findings in the rodent MEC and LEC, respectively (*Khan et al., 2014*).

**eLife digest** In the early 1950s, an American named Henry Molaison underwent an experimental type of brain surgery to treat his severe epilepsy. The surgeon removed a region of the brain known as the temporal lobe from both sides of his brain. After the surgery, Molaison's epilepsy was greatly improved, but he was also left with a profound amnesia, unable to form new memories of recent events.

Subsequent experiments, including many with Molaison himself as a subject, have attempted to identify the roles of the various structures within the temporal lobes. The hippocampus—which is involved in memory and spatial navigation—has received the most attention, but in recent years a region called the entorhinal cortex has also come to the fore. Known as the gateway to the hippocampus, the entorhinal cortex relays sensory information from the outer cortex of the brain to the hippocampus.

In rats and mice the entorhinal cortex can be divided into two subregions that have distinct connections to other parts of the temporal lobe and to the rest of the brain. These are the medial entorhinal cortex, which is the subregion nearest the centre of the brain, and the lateral entorhinal cortex, which is to the left or right of the centre.

For many years researchers had assumed that human entorhinal subregions were located simply to the center or to the sides of the brain. However, it was difficult to check this as the entorhinal cortex measures less than 1 cm across, which placed it beyond the reach of most brain-imaging techniques. Now, two independent groups of researchers have used a technique called functional magnetic resonance imaging to show a different picture. The fMRI data—which were collected in a magnetic field of 7 Tesla, rather than the 1.5 Tesla used in previous experiments—reveal that the entorhinal cortex is predominantly divided from front-to-back in humans.

One of the groups—Navarro Schröder, Haak et al.—used three different sets of functional MRI data to show that the human entorhinal cortex has anterior-lateral and posterior-medial subregions. In one of these experiments, functional MRI was used to measure activity across the whole brain as subjects performed a virtual reality task: this task included some components that involved spatial navigation and other components that did not. The other group—Maass, Berron et al.—used the imaging data to show that the pattern of connections between the anterior-lateral subregion and the hippocampus was different to that between the posterior-medial subregion and the hippocampus.

The discovery of these networks in the temporal lobe in humans will help to bridge the gap between studies of memory in rodents and in humans. Given that the lateral entorhinal cortex is one of the first regions to be affected in Alzheimer's disease, identifying the specific properties and roles of these networks could also provide insights into disease mechanisms.

However, the localization of the human homologue of the rodent MEC and LEC remains unclear. A source of considerable confusion is the fact that 'MEC' and 'LEC' are referring to cytoarchitectonically defined areas and not to anatomical locations. Hence, they do not circumscribe strips of medial and lateral EC. Rather, the MEC is located medially in the septal (posterior) part of the EC and the LEC is located laterally in the temporal (anterior) part of the EC in rodents (*Van Strien et al., 2009*). Furthermore, tracing studies on PHC and PRC pathways in non-human primates suggest a dominant anterior-posterior division (*Suzuki and Amaral, 1994*; *Insausti and Amaral, 2008*), as do single-unit recordings that show activity consistent with the rodent LEC in the anterior EC in primates (*Killian et al., 2012*). In contrast, neuroimaging studies on memory in healthy participants (*Schultz et al., 2012*; *Reagh and Yassa, 2014*) and participants with preclinical AD (*Khan et al., 2014*) suggest that the rodent MEC and LEC map on medial and lateral strips of EC in humans.

To resolve this discrepancy in the literature, one needs to investigate the relatively small EC (25–30 mm$^2$ in humans) (*Krimer et al., 1997*) with high anatomical precision. An earlier study investigated entorhinal connectivity with high-resolution functional magnetic resonance imaging (fMRI), but averaged signal changes over the entire region (*Lacy and Stark, 2012*). To achieve higher resolution imaging, here we leveraged high-field, sub-millimetre fMRI at 7 T and sought to identify the human homologue of the rodent MEC and LEC by informing our analysis by well-known functional and structural properties of the EC. Specifically, it has been shown that MEC and LEC

exhibit differential connectivity with cortical regions (*Witter and Groenewegen, 1989*; *Kerr et al., 2007*; *Van Strien et al., 2009*). The differential fingerprints of anatomical connectivity should lead to differences in functional connectivity identifiable with fMRI (*Johansen-Berg et al., 2004*; *Buckner et al., 2013*; *Wang et al., 2014*). To test patterns of functional connectivity, we measured whole-brain activity while participants performed a virtual reality task with spatial and non-spatial components and validated the results in publicly available resting-state data from the WU-Minn Human Connectome Project (*Van Essen et al., 2013*; *Smith et al., 2013*) (HCP—www.humanconnectome.org). In addition, differential sensitivity to spatial and non-spatial stimuli could provide converging evidence to identify the human homologue of the rodent MEC and LEC, which we tested in a third, independent dataset. A complementary approach to the global network perspective presented here is given by Maass et al. (*Maass et al., 2015*) who scrutinized the fine-grained connectivity pattern of medial temporal lobe regions with the EC.

## Results

A recent model on cortical memory networks (*Ranganath and Ritchey, 2012*) posits that an anterior-temporal (AT) system converges on the PRC and a posterior-medial (PM) system on the PHC. Based on studies in rodents, the two networks are hypothesised to connect to either the LEC or the MEC, respectively (*Witter and Groenewegen, 1989*; *Kerr et al., 2007*; *Van Strien et al., 2009*). Studies in non-human primates predict that the entorhinal projections of the two systems show a strong anteroposterior division (*Suzuki and Amaral, 1994*). In order to test this prediction and to elucidate the role of the EC, we first applied a model-based approach on fMRI data acquired while participants were performing a virtual-reality navigation task to directly mimic studies in rodents (see 'Materials and methods' for details). This task targeted all entorhinal systems, because it involved both navigation-related spatial components and processing of non-spatial stimuli.

We created spherical regions-of-interest (ROIs) with 4 mm radius around coordinates pertaining to either of the networks (*Libby et al., 2012*; *Ranganath and Ritchey, 2012*) (see *Table 1*), as well as ROIs for both the medial and lateral half, and anterior and posterior half of the EC to ensure comparable number of voxels per parcel and therefore comparable signal-to-noise ratio (SNR) properties. Then we computed seed-based connectivity from the two neocortical networks to either sets of EC ROIs, see *Figure 1*. We found a main effect of network on entorhinal connectivity (repeated-measures ANOVA: $F_{(1,21)} = 10.0$, $p = 0.005$). Post-hoc t-tests revealed that the lateral parts of EC connected stronger to the AT compared to the PM network ($T_{(21)} = 2.6$, $p = 0.015$; medial parts of EC: $T_{(21)} = -0.2$ $p = 0.83$). However, in contrast to previous suggestions, we additionally observed a connectivity difference along the anteroposterior axis (repeated-measures ANOVA: main effect of network, $F_{(1,21)} = 13.2$, $p = 0.001$) and post-hoc t-tests showed that the anterior parts of EC connected more with the AT compared to the PM network ($T_{(21)} = 2.7$: $p = 0.01$; posterior EC: $T_{(21)} = -0.49$, $p = 0.63$).

In a second step, we wanted to overcome potential limitations of the seed-based analysis. For example, the selected volume and location of neocortical seed regions could introduce biases (e.g., spatial proximity of the seeds) and imperfect normalisation procedures could affect the results particularly in the frontal lobes where projections from both the rodent LEC and MEC are neighbouring (*Kerr et al., 2007*). In addition, manual subdivision of the EC along cardinal axes likely misrepresents cytoarchitectonic boundaries. Therefore, we adopted a complementary approach to trace the dominant modes of functional connectivity change within the EC in a fully data-driven manner (*Haak et al., 2014*) (see 'Materials and methods'). In brief, for every voxel in the EC, we determined its functional connectivity fingerprint with respect to the rest of cortex and used these fingerprints to compute the pair-wise similarities among all voxels within the ROI. The ensuing (voxels-by-voxels) similarity matrix was then fed to the Laplacian Eigenmaps (LE) algorithm (*Belkin and Niyogi, 2003*), which has previously also been successfully applied to trace changes in white-matter tractography (*Johansen-Berg et al., 2004*; *Cerliani et al., 2012*) and resting-state fMRI connectivity (*Haak et al., 2014*). The LE algorithm projects the high-dimensional, voxel-wise connectivity data onto a series of one-dimensional vectors, with the requirement that the similarities among the connectivity fingerprints are maximally preserved (in the vein of e.g., multidimensional scaling). These vectors represent multiple, spatially overlapping maps (as revealed by colour-coding the EC voxels according to the vectors' values) and are sorted according to how well they preserve the similarities among the original, high-dimensional connectivity fingerprints.

**Table 1.** Selection of regions associated with the posterior-medial (PM) and the anterior-temporal (AT) system (**Libby et al., 2012**).The coordinates of the PM system reflect peak voxel coordinates of a seed-based connectivity contrast of right parahippocampal cortex > right perirhinal cortex connectivity reported by Libby et al. (**Libby et al., 2012**). The coordinates of the AT system reflect peak voxel coordinates of a seed-based connectivity contrast of right perirhinal cortex > right parahippocampal cortex connectivity. Coordinates are in MNI space.

| | Left hemisphere | | | Left hemisphere | | |
|---|---|---|---|---|---|---|
| | x | y | z | x | y | z |
| PM System | | | | | | |
| Medial posterior occipital cortex (BA 18) | – | – | – | 14 | –72 | 8 |
| Occipital pole (BA 17) | –16 | –96 | 22 | – | – | – |
| Parahippocampal cortex | –12 | –42 | –8 | 22 | –32 | –8 |
| Posterior cingulate cortex (BA 29) | –4 | –46 | 4 | 10 | –44 | 10 |
| Posterior hippocampus | –20 | –30 | –2 | 18 | –36 | 0 |
| Posterior thalamus | –20 | –34 | 0 | 22 | –30 | 6 |
| Retrosplenial cortex (BA 30) | –16 | –52 | –4 | 22 | –46 | 0 |
| AT System | | | | | | |
| Dorsolateral prefrontal cortex (BA 9) | –24 | 60 | 24 | 18 | 58 | 24 |
| Dorsomedial prefrontal cortex (BA 8) | –2 | –60 | 34 | – | – | – |
| Frontal polar cortex (BA 10) | – | – | – | 40 | 60 | –2 |
| Lateral precentral gyrus (BA 6) | – | – | – | 54 | 4 | 10 |
| Medial prefrontal cortex (BA 8) | –2 | –60 | 34 | – | – | – |
| Orbitofrontal cortex (BA 11/47) | –6 | 16 | –22 | 8 | 22 | –20 |
| Postcentral gyrus (BA 4) | – | – | – | 62 | –10 | 16 |
| Posterior superior temporal gyrus (BA 22) | –62 | –34 | 14 | – | – | – |
| Rostrolateral prefrontal cortex (BA 10) | – | – | – | 38 | 60 | –12 |
| Temporal polar cortex (BA 38) | – | – | – | 34 | 22 | –36 |
| Ventrolateral prefrontal cortex (BA 44/45) | –56 | 6 | 18 | – | – | – |

Thus, the first vector represents the dominant mode of connectivity change in the EC, the second represents the second-dominant mode, and so on.

Applied to the fMRI data acquired while subjects performed the virtual-reality task, we observed that the dominant mode of functional connectivity change extended along the long-axis of the EC, approximately from the posterior to the anterior end (**Figure 2** and **Figure 2—figure supplement 1**), while the orientation of the second was largely perpendicular (**Figure 3** and **Figure 3—figure supplement 1**). Both modes of connectivity change could also be reliably detected using an independent resting-state fMRI dataset (60 subjects of the WU-Minn Human Connectome Project; see 'Materials and methods'), suggesting that the organization of EC functional connectivity is largely task-independent (**Figure 4**). Both the first and second-dominant modes of functional connectivity change were highly reproducible across resting-state sessions (Pearson's R = 0.99, p < 0.001 and Pearson's R = 0.98, p < 0.001, for the dominant and second-dominant modes, respectively).

Furthermore, in order to identify the potential human homologues of the rodent LEC and MEC, we clustered the vectors representing the dominant and second-dominant modes of functional connectivity change (separately) through a median-split approach (see 'Materials and methods'). Hence, each cluster comprises 50% of voxels in the EC. 3D-rendering of the two clusters derived from the dominant mode of connectivity change revealed a consistent topology across hemispheres (**Figure 2B**, **Video 1**). One division contained the posterior EC (**Figure 2—figure supplement 1**—displayed in blue). The other division included most of the anterior EC (**Figure 2—figure supplement 1**—displayed in red). In addition to the dominant anterior-posterior distinction, the posterior cluster was located more

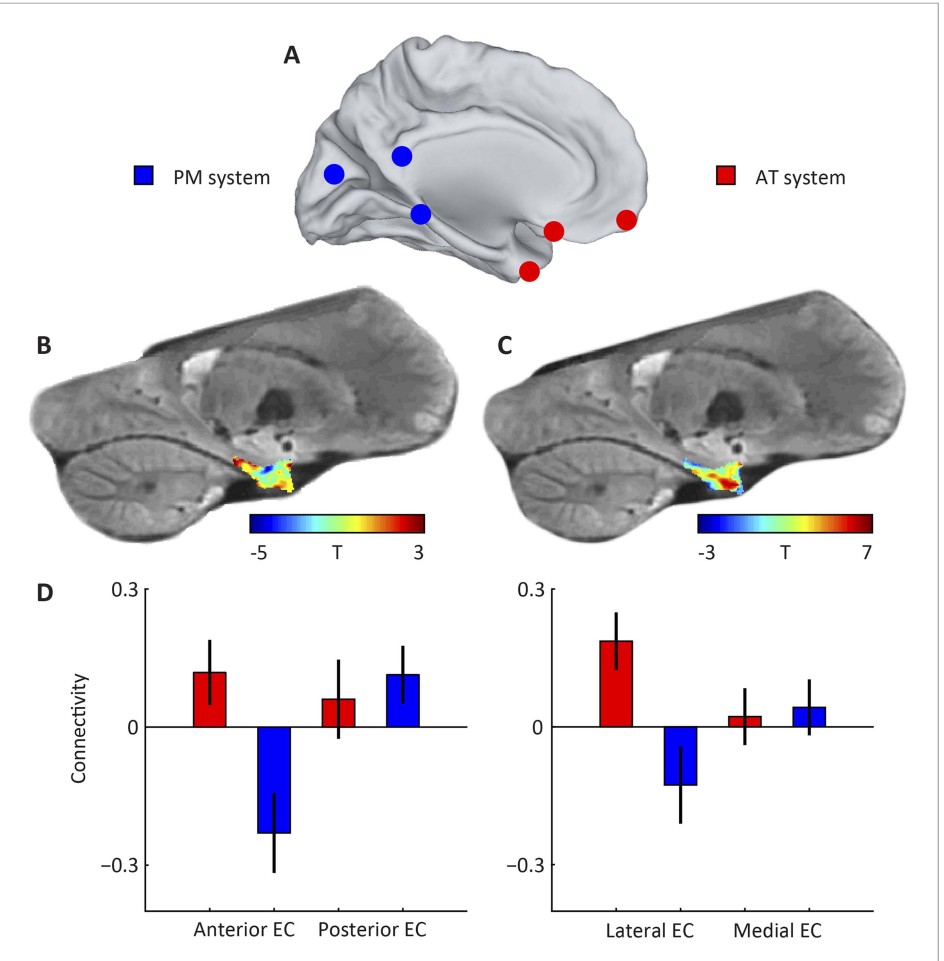

**Figure 1**. Subdivisions of entorhinal cortex (EC) and connectivity to anterior-temporal (AT) and posterior-medial (PM) cortical networks. (**A**) Schematic of the AT and PM system. Spherical regions-of-interest (ROIs) were centred on MNI coordinates associated with either of the two systems (**Libby et al., 2012**), normalised to the group-specific template of the navigation study and then masked to include only gray matter voxels. The AT system included medial-prefrontal and orbitofrontal regions, whereas the PM system included occipital and posterior-parietal regions, see **Table 1** for all selected regions. (**B**) Right parasagittal slice showing voxel-wise seed-based connectivity of the PM system restricted to the EC. Note the PM peak. (**C**) Right parasagittal slice showing voxel-wise seed-based connectivity of the AT system restricted to the EC. Note a peak in the anterior-lateral EC. (**D**) ROI-based connectivity estimates. Left panel: Connectivity strength (partial correlation coefficient) of anterior (left) and posterior EC (right) is plotted separately for the AT system (red) and the PM system (blue). The systems differ in their entorhinal connectivity: the anterior EC connects stronger to the AT compared to the PM network. Right panel: Connectivity strength with lateral (left) and posterior EC (right) is plotted separately for the AT system (red) and the PM system (blue). Lateral EC connected stronger to the AT compared to the PM network. Error bars show S.E.M. over subjects. See **Figure 1—figure supplement 1** for additional slices.

The following figure supplement is available for figure 1:

**Figure supplement 1**. Results of the model-based connectivity analyses (additional slices).

medially (and to some extent more dorsally) and the anterior cluster was located more laterally (and to some extent more ventrally). Hereafter, we refer to the clusters as posterior-medial EC ('pmEC') and anterior-lateral EC ('alEC'), respectively, consistent with Maass et al. (**Maass et al., 2015**). Clusters derived from the second-dominant mode were less consistent across hemi-spheres, but showed an approximately orthogonal orientation relative to the first (**Figure 3** and **Figure 3—figure supplement 1**).

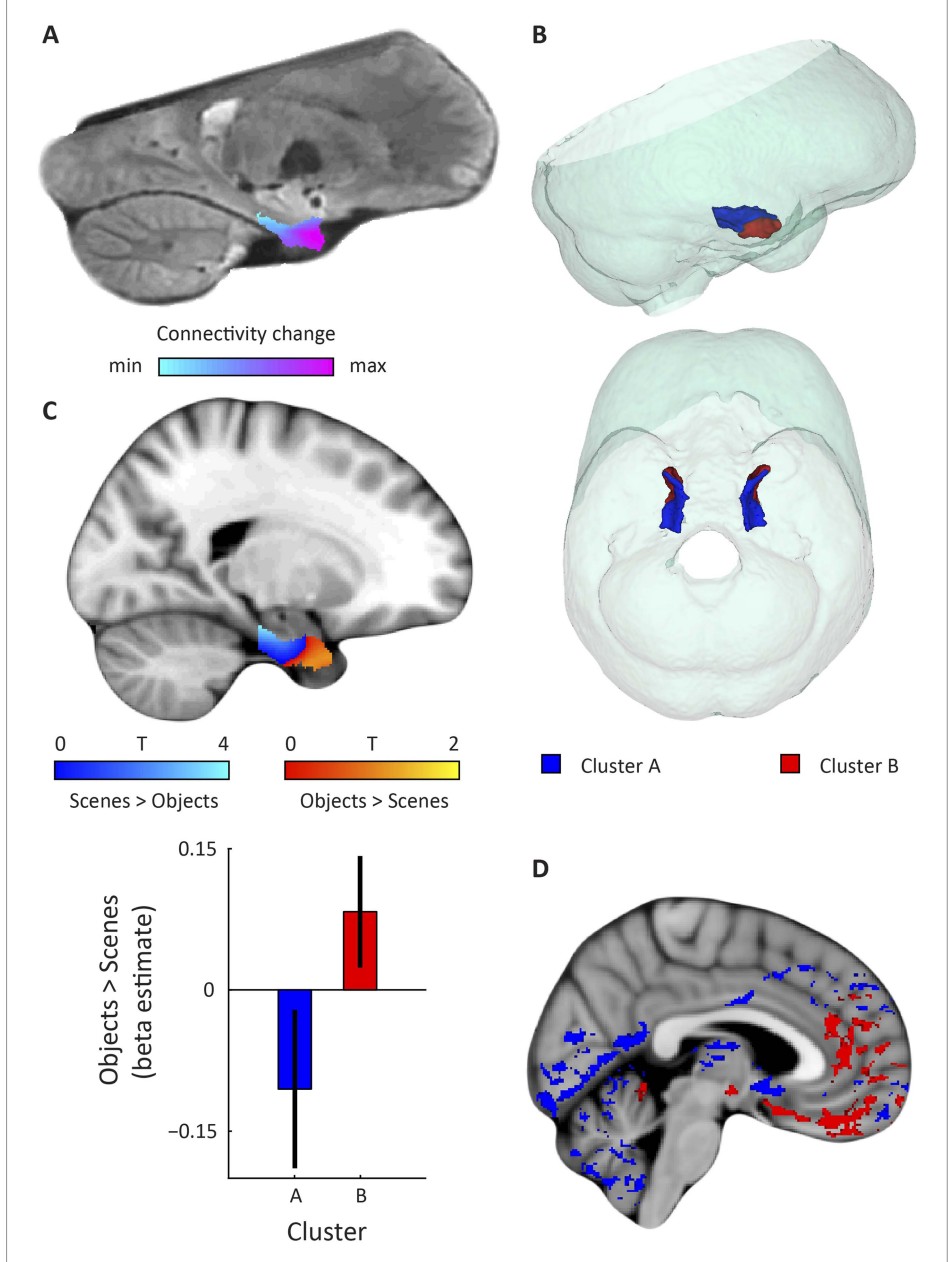

**Figure 2**. Dominant mode of functional connectivity change within EC and sensitivity to spatial and non-spatial information. (**A**) Dominant mode of functional connectivity change at the group-level (Spearman's R = 0.53). Similar colours indicate similar connectivity with the rest of the brain. (**B**) 3D rendering of the two clusters derived from the dominant mode of functional connectivity change (displayed in red and blue) and the outlines of the group-specific template. Upper panel: right side view. Lower panel: top view (see *Figure 2—figure supplement 1* for coronal views of the two clusters). (**C**) Upper panel: Map shows results of a non-parametric randomisation test of the spatial and non-spatial stimulation experiment restricted to the EC for display purposes (see *Figure 2—figure supplement 2* for whole-brain maps). The 'scenes > objects' contrast is displayed in blue to light-blue, the 'scenes < objects' contrast in red to yellow. Note that voxels in pmEC are sensitive to scenes, whereas voxels in alEC are sensitive to objects. Lower panel: The clusters from panel B exhibit antagonistic responses to spatial and non-spatial stimuli. Beta estimates for the contrast 'scenes > objects' (averaged across participants) are shown for clusters A and B. $T_{(20)} = 4.9$, p = 0.0001. Error bars show S.E.M. over participants. (**D**) Whole-volume functional connectivity with clusters A and B. Regions connecting more with cluster A (p < 0.05, FWE corrected), such as occipital and

*Figure 2. continued on next page*

*Figure 2. Continued*

posterior-parietal cortex that form part of the PM system are shown in blue. Regions connecting more with cluster B (p < 0.05, FWE corrected), such as medial-prefrontal and orbitofrontal cortex which form part of the AT system are displayed in red.

The following figure supplements are available for figure 2:

**Figure supplement 1**. Coronal views of the two clusters.

**Figure supplement 2**. Whole-brain modulation by spatial and non-spatial stimuli.

**Figure supplement 3**. Signal-to-noise ratios (SNRs) in the alEC and pmEC.

**Figure supplement 4**. Homologous and non-homologous connectivity of the alEC and the pmEC.

If the two clusters (i.e., EC halves) derived from the dominant mode of functional connectivity change correspond to the homologues of the rodent LEC and MEC, their whole-brain connectivity profiles should correspond to the known connectivity profiles in rodents and resemble the AT and PM system proposed by Ranganath and Ritchey (*Ranganath and Ritchey, 2012*). To test this hypothesis, we computed whole-volume connectivity maps of the two clusters. Group-level contrasts showed peaks in the medial-prefrontal and orbitofrontal cortex for the alEC, regions associated with the AT system. In contrast, occipital and posterior-parietal cortex was dominated by connectivity with the pmEC, areas associated with the PM system (*Ranganath and Ritchey, 2012*). In addition, the pmEC showed increased connectivity with frontal regions (see *Figure 2D*). These findings are in line with the patterns of reciprocal connections of the rodent LEC and MEC, respectively (*Kerr et al., 2007*). Notably, this was not the case for the connectivity maps of the two clusters derived from the second-dominant mode of connectivity change (*Figure 3D*) that were widely dominated by only one of the clusters.

Furthermore, we examined both spatial and temporal SNR (tSNR) of the alEC and the pmEC (*Figure 2—figure supplement 3*). tSNR did not differ between alEC and pmEC ($T_{(21)} = 0.2$, $p = 0.83$) but spatial SNR did ($T_{(21)} = 9.7$, $p < 0.001$). This was associated with higher signal in the pmEC compared to the alEC (mean signal: alEC = 4.97; mean signal pmEC = 7.76; $T_{(21)} = 38$, $p < 0.001$) in the absence of differences in spatial standard deviation ($T_{(21)} = 1.5$, $p = 0.144$). Note, that mean signal was subtracted from time-series prior to all connectivity analyses (see 'Materials and methods'), which makes it unlikely that signal intensity differences affected the connectivity results.

We used an independent component analysis (ICA)-based method for data cleaning (see 'Materials and methods') that has been shown to efficiently remove residual effects of head motion. However, we additionally repeated the data-driven connectivity analysis after excluding time periods with large head movements (motion scrubbing [*Power et al., 2012*, *2015*]). Motion scrubbing had only minimal effects on the results. Pearson correlation coefficients of the pre- and post scrubbing results were close to 1 and highly significant (gradient one: left R = 0.9964, right R = 0.9958; gradient two: left = 0.9348, right = 0.8816; all p values <0.001).

We also tested if the alEC and pmEC exhibited stronger connectivity with their potential homologue region in the contralateral hemisphere compared to each other. Here we observed that homologous connectivity indeed exceeded non-homologous connectivity (*Figure 2—figure supplement 4*). Importantly, this was also the case if non-homologous connectivity was assessed within the same hemisphere, between adjacent parts of the EC ($T_{(21)} = 4.05$, $p = 0.0006$).

Finally, studies on rodent electrophysiology (*Deshmukh and Knierim, 2011*; *Knierim et al., 2013*) predict that the LEC and its human homologue should respond preferentially to non-spatial stimuli, whereas the MEC and its human homologue should be involved in processing spatial information. We tested this prediction by conducting a second fMRI study at 7 T in which an independent group of participants was presented with spatial (pictures of scenes) and non-spatial stimuli (pictures of objects), see 'Materials and methods' for details. We contrasted fMRI responses to spatial and non-spatial stimuli. Here, we observed higher responses to spatial than non-spatial stimuli in the posterior EC, while the inverse contrast (objects vs scenes) showed higher responses in the anterior EC (*Figure 2C*).

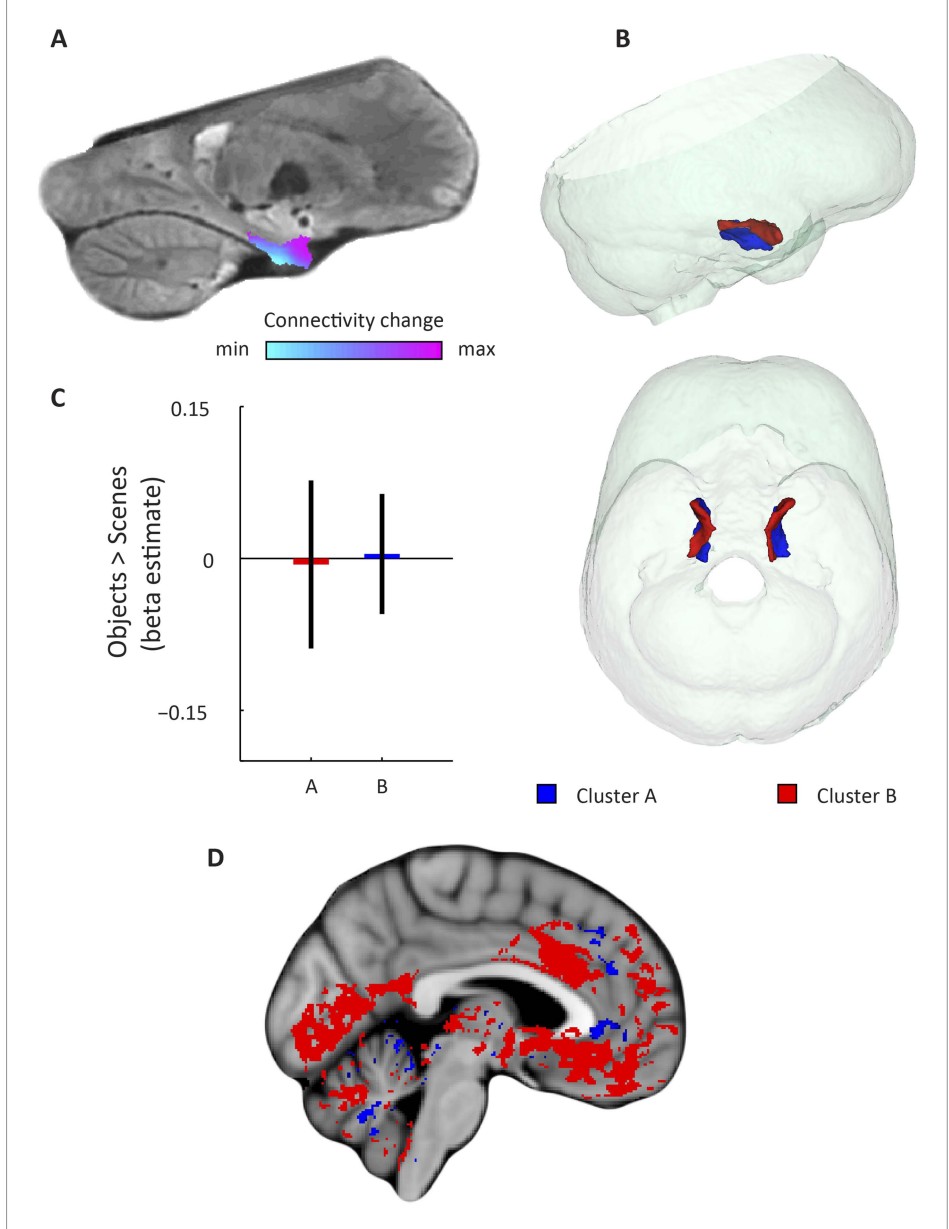

**Figure 3**. Second-dominant mode of functional connectivity change within EC and sensitivity to spatial and non-spatial information. (**A**) Second-dominant mode of functional connectivity change at the group-level (Spearman's R = 0.28). Similar colours indicate similar connectivity with the rest of the brain. (**B**) 3D rendering of the two clusters derived from the second-dominant mode of functional connectivity change (displayed in red and blue) and the outlines of the group-specific template. Upper panel: right side view. Lower panel: top view (see *Figure 3—figure supplement 1* for coronal views of the two clusters). (**C**) The clusters shown in panel B exhibit no antagonistic responses to spatial and non-spatial stimuli. Beta estimates for the contrast 'scenes > objects' (averaged across participants) are shown for cluster A and B ($T_{(20)} = -0.26$, $p = 0.8$). Error bars show S.E.M. over participants. (**D**) Regions connecting more with cluster A ($p < 0.05$, FWE corrected) are shown in blue. Regions connecting more with cluster B ($p < 0.05$, FWE corrected) are shown in red. Cluster A connected more with most of the neocortex.

The following figure supplement is available for figure 3:

**Figure supplement 1**. Coronal views of the two clusters.

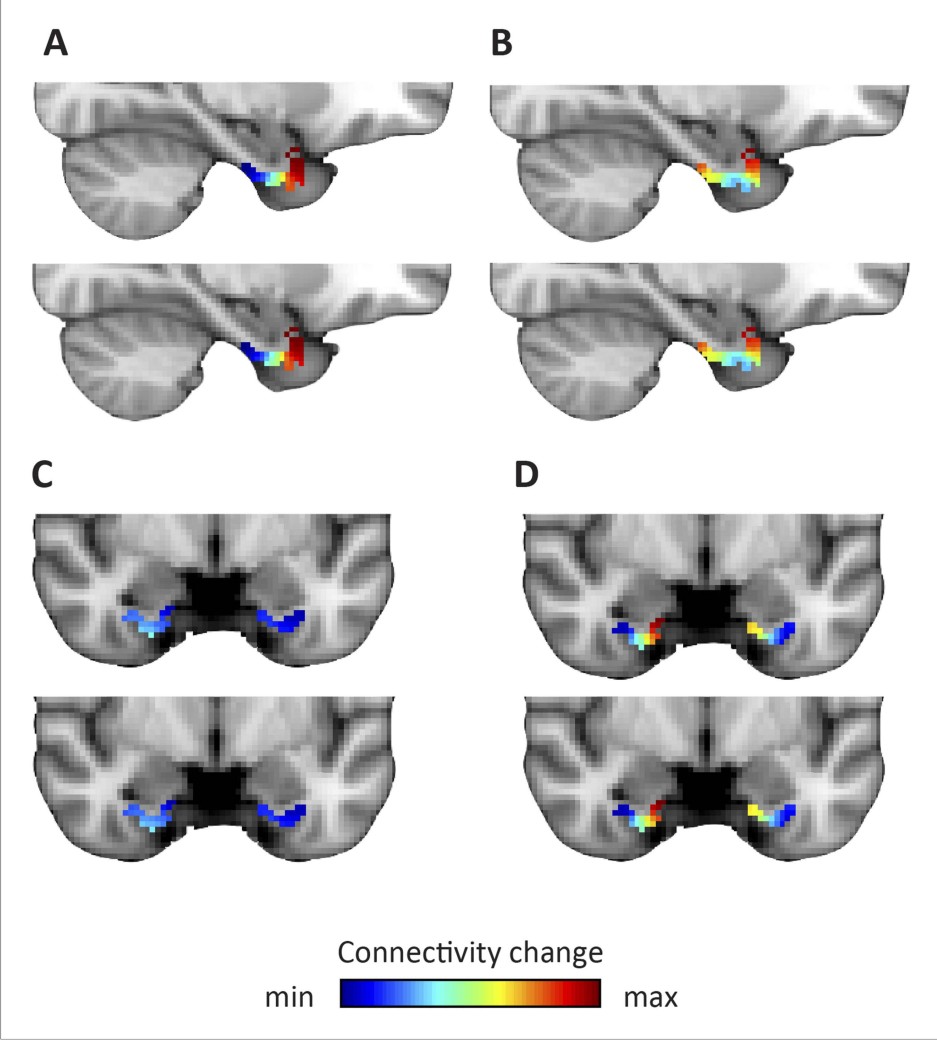

**Figure 4**. Dominant and second-dominant modes of functional connectivity change on the basis of resting-state functional magnetic resonance imaging. Results of analysis of the first 60 participants of the WU-Minn Human Connectome Project (HCP), acquired on two different days (*Smith et al., 2013*). Top row: day one. Bottom row: day two. (**A**, **C**) The dominant mode of functional connectivity change follows an anteroposterior trajectory. (**B**, **D**). The second-dominant mode of functional connectivity change follows a mediolateral trajectory. Both modes were highly reproducible across different scanning days (dominant mode: Pearson's R = 0.99 p < 0.001, second-dominant mode: R = 0.98; p < 0.001). Topology preservation—dominant mode, day one: Spearman's R = 0.62; day two: R = 0.61; second-dominant mode, day one: R = 0.44; day two: R = 0.46.

ROI analyses using the clusters derived from the dominant mode of connectivity change revealed that the anterior-lateral cluster showed higher sensitivity to non-spatial stimuli compared to the PM cluster ($T_{(20)}$ = 4.9, p = 0.0001, *Figure 2C*). This dissociation was not present for the clusters that were derived from the second-dominant mode of connectivity change within the EC ($T_{(20)}$ = −0.26, p = 0.8; *Figure 3C*).

In sum, our results suggest that the human homologue of the rodent MEC maps predominantly on the human posterior parts of the EC, while the homologue of the rodent LEC maps predominantly on the anterior parts of the EC.

## Discussion

The EC, in concert with the hippocampus, plays a crucial role in memory and learning (*Eichenbaum et al., 2007*) and is the core of the brain's navigational system (*Moser and Moser, 2013*). While the

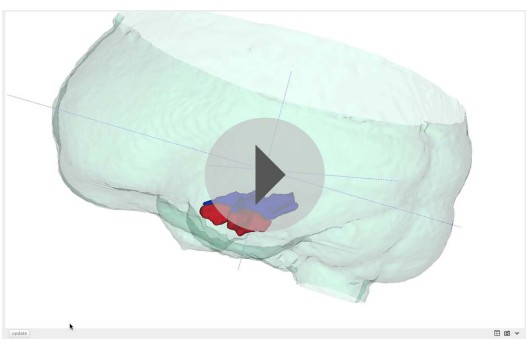

**Video 1.** 3D rendering of the two clusters derived using the dominant mode of functional connectivity change. Cluster A is shown in red and cluster B in blue.

shape and location of the EC differs between rodents and primates (*Witter and Groenewegen, 1989*), the anatomical organisation and connectivity patterns are largely conserved across species (*Canto et al., 2008*). However, translational studies on EC subregions faced the problem of identifying homologous regions across species. For example, recent neuroimaging studies on mnemonic processing (*Schultz et al., 2012*; *Reagh and Yassa, 2014*) and Alzheimer's pathology (*Khan et al., 2014*) directly related medial and lateral strips of EC in humans to the rodent MEC and LEC. However, the anatomical locations of these cytoarchitectonically defined regions in rodents differ along more than only the mediolateral axis. More specifically, the LEC is situated more anterior-ventrally, whereas the MEC is situated more posterior-dorsally in rodents (*Van Strien et al., 2009*). Therefore, it is unlikely that medial and lateral strips of EC in humans correspond to the rodent MEC and LEC, respectively. Furthermore, in primates the characteristic projections from the PRC and PHC strongly map onto the anteroposterior axis (*Suzuki and Amaral, 1994*). Here, we leveraged the distinct connectivity fingerprints and functional roles (such as complementary computation of scene and object information) of the rodent LEC and MEC to find their human homologues with fMRI with three complementary methods and three independent datasets. Both model-based and data-driven connectivity analyses, as well as sensitivity to non-spatial vs spatial stimuli provide evidence for an anterior-lateral and a PM localisation of the homologues of the rodent LEC and MEC, respectively. Maass et al confirmed these findings in a study with two high-resolution, high-field fMRI datasets by focusing on local connectivity between regions of the medial temporal lobes. They found preferential connectivity of PRC and proximal subiculum to anterior-lateral parts of the EC, whereas posterior-medial parts of the EC were more connected to PHC and distal subiculum. This corresponds well with our findings (*Figure 2—figure supplement 1*). In line with the present study, Maass et al (*Maass et al., 2015*) report local connectivity fingerprints of the human anterior-lateral and posterior-medial EC that mimicked those of the rodent LEC and MEC, respectively.

In addition to the change in functional connectivity from PM to anterior-lateral, our data-driven connectivity analysis also revealed a second organisation structure approximately perpendicular to the first (*Figure 3—figure supplement 1*), which might reflect bands of intra-entorhinal projections that are known to cross the LEC/MEC boundary in a roughly orthogonal orientation in rodents (*Canto et al., 2008*) and in primates (*Chrobak and Amaral, 2007*).

The selective sensitivity to spatial and non-spatial information, or 'context vs content' more broadly (*Knierim et al., 2013*), points towards fundamental difference in computations of the LEC and MEC. How to characterise those differences most accurately remains an open question (*Knierim et al., 2013*), but our results can help to inform future studies on the role of the human alEC and pmEC in higher-level cognition.

Notably, the present findings confirm three out of four complementary criteria for the definition of cortical areas that have traditionally been advocated (*Van Essen, 1985*), namely topographic organization, connectivity and functional properties (the fourth one being cyto- and myeloarchitectonic organization).

Previous neuroimaging studies in humans reported differences between medial and lateral aspects of EC that mimicked differences between the rodent MEC and LEC and assumed that both subregions are present on coronal slices of the EC (*Schultz et al., 2012*; *Khan et al., 2014*; *Reagh and Yassa, 2014*), that is, that the MEC and LEC correspond to medial and lateral strips of the EC. In light of our findings, these reports could be explained by a partial overlap of the medial and lateral divisions with the pmEC and the alEC, respectively. For example, we noticed a mediolateral difference of responses to spatial and non-spatial stimuli on some coronal slices (*Figure 2—figure supplement 2B*). However, our results suggest that coronal slices through the most posterior EC exclusively harbour the human homologue of the rodent MEC. Similarly, anterior slices appear to contain mostly the homologue

of the rodent LEC. Hence, improved mapping of homologous regions between rodents and humans should lead to increased effect sizes and more accurate interpretations.

In summary, the present findings can help to inform future translational research on the role of entorhinal subregions in fields ranging from clinical neuroscience, such as on the early progression of Alzheimer's disease, to cognitive neuroscience, for example, nature and mechanisms of different forms of memory and their integration into higher order representations (*Eichenbaum and Lipton, 2008*).

# Materials and methods

## Participants

### Navigation experiment
26 participants took part in the study (11 females, age 19–36, mean 23 years). 22 entered the analysis and 4 were excluded due to excessive movement (number of instantaneous movements [*Power et al., 2012*] > 0.5 mm exceeded the mean plus 1 standard deviation). Materials and methods were approved by the local research ethics committee (CMO University Duisburg-Essen, Germany and CMO region Arnhem-Nijmegen, NL). Written informed consent was obtained from each participant for data analysis and publication of the study results.

### Resting-state dataset
The resting-state dataset consisted of the first 60 participants (41 females, age ranges: 4 between 22 and 25 years, 23 between 26 and 30 years and 33 between 31 and 35 years) of the WU-Minn Human Connectome Project (HCP) (*Van Essen et al., 2013*). The experiments were performed in accordance with relevant guidelines and regulations and all experimental protocol was approved by the Institutional Review Board (IRB) (IRB # 201204036; Title: 'Mapping the Human Connectome: Structure, Function, and Heritability'). Written informed consent was obtained from each participant for data analysis and publication of the study results.

### Spatial and non-spatial stimulation
21 participants (13 females, age 20–54, mean age 26 years) participated in the experiment. None of the participants participated in more than one experiment. Materials and methods were approved by the local research ethics committee (CMO University Duisburg-Essen, Germany and CMO region Arnhem-Nijmegen, NL). Written informed consent was obtained from each participant for data analysis and publication of the study results.

## FMRI acquisition

### Navigation experiment
Blood-oxygenation-level-dependent T2*-weighted functional images were acquired on a 7 T Siemens MAGNETOM scanner (Siemens Healthcare, Erlangen, Germany) using a three dimensional echo-planar imaging (3D EPI) pulse sequence (*Poser et al., 2010*) on a 32-channel surface coil: TR = 2.7 s, TE = 20 ms, flip angle = 14˚, slice thickness = 0.92 mm, slice oversampling = 8.3%, in-plane resolution = 0.9^2 mm, field of view (FoV) = 210 mm in each direction, 96 slices, phase encoding acceleration factor = 4, 3D acceleration factor = 2. The first five volumes of the main scan were discarded to allow for T1 equilibration. A field map using a gradient echo sequence was recorded for distortion correction of the acquired EPI images.

### Resting-state dataset
Participants underwent four resting-state scanning sessions recorded on two days, each comprising 15 min multi-band accelerated (TR = 0.72 s) fMRI. The two scans from each day were concatenated into scans of 30 min. Each day one scan had a left-right phase encoding direction and the other a right-left phase encoding direction and concatenating the two scans allowed for correcting the ensuing opposing field inhomogeneities. Whole-brain images were acquired at 2 mm isotropic resolution and all data are publicly available. For details see Van Essen et al (*Van Essen et al., 2013*; *Smith et al., 2013*).

### Spatial and non-spatial stimulation
The same 7 T Siemens MAGNETOM scanner was used as for the navigation experiment. Procedures and acquisition parameters were the same unless described otherwise. TR = 2.1 s, TE = 25 ms,

flip angle = 13°, slice oversampling = 10%, in-plane resolution = 1.5^2 mm, FoV = 224 mm in each direction, 80 slices, Phase Encoding acceleration factor = 3, 3D acceleration factor = 2. To allow for T1 equilibration, three (N = 11) or seven (N = 5) dummy volumes were discarded before the main scan.

## Experimental tasks

### Navigation experiment

Participants performed an object-location memory task while freely navigating a 3D, virtual environment task adapted from Doeller et al. (*Doeller et al, 2008*; *Doeller et al., 2010*) (*Figure 5A*). In this self-paced task participants collected and replaced six everyday objects within a virtual arena. They collected each object from its associated location once during an initial trial, by running over it. In each subsequent trial they saw an image (cue) of one of the objects in the upper part of the screen and had to move to the object's associated location and press a button (replace phase). After this response, the object appeared in its associated position and participants collected it again (feedback phase). After an average of 3 trials (range 2–4 trials), a fixation cross on a grey background was presented for 4 s (inter-trial-interval, ITI). Next to the spatial aspects of the navigation task, non-spatial task components were present approximately half of the time. Therefore, the task engaged functions associated with both the PM and the AT network, respectively. Each scanning session was subdivided into EPI acquisition blocks of 210 volumes. Participants underwent an average of 5 blocks (±1). Task-related effects from the navigation experiment are subject of another report.

### Spatial and non-spatial stimulation

Participants were presented with pictures of spatial and non-spatial stimuli (see *Figure 5B*) and gave a trial-by-trial animacy judgement. Images of objects (non-spatial) and 12 images of scenes (spatial)—as well as 24 images of two categories of animate stimuli not considered in the analyses—were resized to 400 × 400 pixels and shown in grey shading and matched on their low-level features (luminance, contrast and spatial frequency) through the use of the SHINE toolbox (Spectrum, Histogram and Intensity Normalization and Equalization [*Willenbockel et al., 2010*]). Single stimuli were presented for 2 s followed by an ITI of variable duration (mean 5 s, range 1.5–10 s). The scanning sessions were subdivided into two runs of four blocks each, with a duration of approximately 27 min in duration per run.

## Data pre-processing

### Navigation experiment

Data pre-processing was implemented through the use of the automatic analysis library (*Cusack et al., 2015*) (http://automaticanalysis.org/). Pre-processing included motion correction in SPM8 (http://www.fil.ion.ucl.ac.uk/spm), data de-noising with the FIX artifact removal procedure implemented in FSL 5.0.4 (http://fsl.fmrib.ox.ac.uk/fsl/) that has been trained manually on 10 of the 22 participants and nonlinear normalisation to a group-specific EPI template with the Advanced Neuroimaging Toolbox (*Avants*

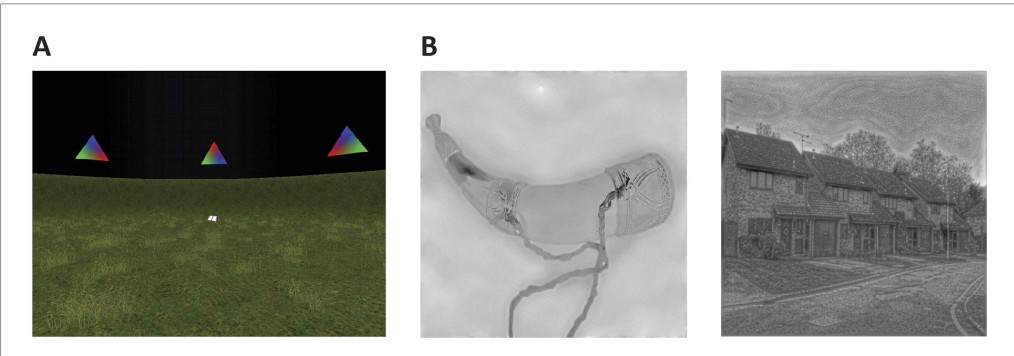

**Figure 5**. Cognitive tasks. (**A**) Navigation experiment. First person view of the virtual arena that participants navigated freely to perform the object-location memory task. (**B**) Spatial and non-spatial stimulation. Left: An example stimulus of a 'non-spatial' object. Right: An example stimulus of a 'spatial' scene.

et al., 2011) (ANTS; http://www.picsl.upenn.edu/ANTS/). The use of a group-specific EPI template was feasible because of the anatomical detail provided by the sub-millimetre images. In addition, this rendered the registration to structural images unnecessary, which poses an additional source of noise because of regional differences in distortion of the 7 Tesla EPIs. The functional data were smoothed with a full-width at half-maximum (FWHM) Gaussian kernel of 2.5 mm³ (roughly 2.5 times the voxel size) to increase the SNR and improve ICA-based data de-noising with the FIX procedure (see above) while maintaining high spatial resolution for connectivity analyses. Correction for residual motion artefacts was performed with an ICA-based method, which has been shown to outperform motion scrubbing and spike-regression methods both in terms of reproducibility of resting-state networks and conservation of temporal degrees of freedom (Pruim et al., 2015a; Pruim et al., 2015b). In addition, brain extraction, tissue segmentation and high-pass filtering with a 128-s cut-off were carried out with FSL.

### Resting-state dataset

The publicly available data were pre-processed as described in Smith et al. (2013), which included correction for spatial distortions and head motion, registration to the T1 weighted structural image, resampling to 2 mm Montreal Neurological Institute (MNI) space, global intensity normalization, high-pass filtering with a cut-off at 2000s, and the FIX artefact removal procedure. Additionally, we smoothed the images with a 6 mm³ FWHM kernel and converted the time-series of each voxel to percent signal change by subtracting and dividing by their mean amplitude value over time.

### Spatial and non-spatial stimulation

The same pre-processing procedure was used as for the navigation experiment, unless described otherwise below. Since the voxel size was larger and only univariate analyses were conducted, we did not perform data de-noising with the FIX procedure. The functional data were smoothed with a FWHM kernel of 5 mm³ (roughly 2.5 times the voxel size) to increase the SNR.

## ROI definitions

For the 7 Tesla experiments (navigation experiment and spatial and non-spatial stimulation experiment), an EC ROI was manually defined with ITK Snap 3.2 (Yushkevich et al., 2006) (http://www.itksnap.org) on the group-specific high-resolution mean EPI template from the navigation experiment. Manual segmentation on the group-specific EPI template circumvented registration problems between structural images and partly distorted functional images. Based on anatomical landmarks as described by Insausti et al., (1998) and Frankó et al., (2014), the posterior border of the EC was set to ~1.5 mm posterior to the gyrus intralimbicus, the anterior border to ~1.5 mm posterior to the limen insulae, the lateral border to the midpoint of the medial bank of the collateral sulcus and the medial border to the hippocampal fissure at the level of the uncus. A medial EC and a lateral EC ROI of roughly equal volume were created by dividing the main ROI mediolaterally (which roughly corresponded to a proximo-distal division in relation to the hippocampus) on consecutive coronal slices along the entire anteroposterior axis. An anterior EC and a posterior EC ROI of roughly equal volume were created by dividing the main ROI mid-way between the most anterior and the most posterior coronal slice of the EC.

For the 'model-based' analysis, one ROI mask was created for the PM system and another one for the AT system. For this purpose we placed spheres of 4 mm radius on MNI coordinates associated with each system by Libby et al., (Libby et al., 2012) (see Table 1 for the selected coordinates). For the control analysis using the resting-state fMRI data from the Human Connectome Project, a probabilistic entorhinal ROI was generated using the Freesurfer toolbox (Augustinack et al., 2013). Freesurfer's cortical reconstruction algorithm ('recon-all') was used on a T1 weighted MNI template image to generate an ROI mask of the EC with a probability threshold of 90% that was binarized and converted to NiFTI file format.

## Seed-based connectivity analysis

For the quantification of the model-based analyses of connectivity between the AT and PM networks and the manually segmented medial and lateral or anterior and posterior EC, singular-value decomposition (SVD) and subtraction of the mean was performed on the voxel-wise time-series within each ROI. Next, functional connectivity was estimated by means of partial correlation analysis between the time-series of groups of ROIs. Finally, Pearson's partial correlation coefficients were Fisher-Z transformed and used for across-subject comparisons (Figure 1D).

For the estimation of voxel-wise connectivity maps we used the 'dual_regression' function implemented in FSL 5.0.4 (http://fsl.fmrib.ox.ac.uk/fsl/). This involved using the time-series of pairs of seed regions as regressors in a GLM to estimate functional connectivity to voxels in a target region (*Figure 1B*, *Figure 1C* and *Figure 1—figure supplement 1*) or to the rest of the brain (*Figure 2D* and *Figure 3D*). Finally, group-level statistics were computed with non-parametric randomisation tests (FSL 5.0.4, http://fsl.fmrib.ox.ac.uk/fsl/) and threshold-free cluster enhancement for null-hypothesis testing and the computation of p-value maps (*Figure 2D* and *Figure 3D*). All connectivity analyses were restricted to gray matter voxels.

## Data-driven connectivity analysis

To overcome limitations of the seed-based connectivity analysis, we employed *ConGrads* (*Haak et al., 2014*), which allows for tracing the dominant modes of functional connectivity change within a pre-specified region of the brain in a fully data-driven manner. First, the fMRI time series from the EC were rearranged into a time-by-voxels matrix, which was also done for the fMRI time series of all gray-matter voxels outside the EC. For reasons of stability and computational tractability, we losslessly reduced the dimensionality of the data outside the EC using SVD. We determined the connectivity fingerprints of each voxel inside the EC by computing the correlation between the voxel-wise time series and the SVD-transformed data, and then used the $\eta^2$ coefficient to quantify the similarities among the voxel-wise fingerprints (*Cohen et al., 2008*). Next, we fed the ensuing similarity-matrix to the LE algorithm (*Belkin and Niyogi, 2003*), resulting in a series of vectors that represent the dominant modes of functional connectivity change. The LE algorithm and variants thereof have previously been successfully applied to trace changes in probabilistic tractography connectivity (*Johansen-Berg et al., 2004*; *Cerliani et al., 2012*), while in the context of resting-state fMRI, *ConGrads* has been shown to generate highly reproducible results in regions such as the human motor strip, both across sessions and participants (*Haak et al., 2014*). Note that group-level results were obtained by running the LE algorithm on the average of the individual similarity matrices and that the analysis was performed separately for left and right hemispheric EC ROIs.

To quantify how well the ensuing modes of connectivity change preserved the order of the similarities among the original, high-dimensional connectivity fingerprints (topology preservation) we used Spearman's rank correlation coefficient. Because the LE algorithm maximizes topology preservation, we report the correlation coefficients without p-values.

Clustering was performed on the high-resolution data from the navigation experiment by grouping voxels above and below the median value of each mode of functional connectivity change, which resulted in two equally sized clusters per mode of connectivity change. For the ROI analyses on the data of the spatial and non-spatial stimulation experiment, the clusters were warped into MNI space.

## SNR estimation

tSNR was determined by dividing the mean signal within a region by the standard deviation of that signal over time. Conversely, spatial SNR was determined by dividing the mean signal within a region by the standard deviation of the signal across voxels. This was done for each time point and spatial SNR was then averaged over time.

## Motion scrubbing

First, we determined time points where instantaneous movement (*Power et al., 2012*, *2015*) exceeded a threshold of 0.5 mm. Then we excluded these time points (volumes) including the one preceding and the two thereafter from subsequent analyses. Hence, per instantaneous movement above threshold, four volumes were removed. On average this resulted in 121 volumes being removed from participant's time-series (range: 4–300).

## Univariate fMRI analyses

The data from the spatial and non-spatial stimulation experiment were analysed in native (subject-specific) space with general linear models that included regressors of interest for object and scene trials. Six movement regressors were included to account for movement-related noise. The regressors were convolved with the canonical hemodynamic response function and

fitted to the time-series at each voxel in a whole-brain analysis. Single-subject contrast images were first normalised to a group-specific template and then to the MNI space. Group-level statistics were computed with non-parametric randomisation tests (FSL 5.0.4, http://fsl.fmrib.ox. ac.uk/fsl/) on the contrast images (objects vs scenes and scenes vs objects) using variance smoothing with a 5 mm$^3$ kernel—to improve the estimation of the variance that feeds into the final t-statistic.

## Acknowledgements

This work was supported by fellowships from the European Research Council (ERC-StG 261177) and the Netherlands Organisation for Scientific Research (NWO-Vidi 452-12-009) awarded to CFD. CFB is supported by the Netherlands Organisation for Scientific Research (NWO-Vidi 864-12-003) and CFB and KVH gratefully acknowledge funding from the Wellcome Trust UK Strategic Award (098369/Z/12/Z). The authors would like to thank A. Vicente-Grabovetsky for help with data analyses, L. Schürmann for providing data, A. Backus for help with data analysis and creation of figures and J. Bellmund, S. Collin and B. Milivojevic for useful discussions and comments on earlier versions of the manuscript.

## Additional information

### Funding

| Funder | Grant reference | Author |
| --- | --- | --- |
| European Research Council (ERC) | ERC-StG 261177 | Tobias Navarro Schröder, Christian F Doeller |
| Nederlandse Organisatie voor Wetenschappelijk Onderzoek | NWO-Vidi 452-12-009 | Tobias Navarro Schröder, Christian F Doeller |
| Nederlandse Organisatie voor Wetenschappelijk Onderzoek | NWO-Vidi 864-12-003 | Christian F Beckmann |
| Wellcome Trust | UK Strategic Award 098369/Z/12/Z | Koen V Haak, Christian F Beckmann |

The funders had no role in study design, data collection and interpretation, or the decision to submit the work for publication.

### Author contributions

TNS, Conception and design, Acquisition of data, Analysis and interpretation of data, Drafting or revising the article; KVH, NIZJ, Analysis and interpretation of data, Drafting or revising the article; CFB, CFD, Conception and design, Drafting or revising the article

### Author ORCIDs

Tobias Navarro Schröder, http://orcid.org/0000-0001-6498-1846

### Ethics

Human subjects: All studies were approved by the local Research Ethics Committees (CMO University Duisburg-Essen, Germany and CMO region Arnhem-Nijmegen, NL and Institutional Review Board 'IRB # 201204036'; Title: 'Mapping the HumanConnectome: Structure, Function, and Heritability', US). Written informed consent was obtained from each participant for data analysis and publication of the study results.

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
