## [Decision Letter]

Thank you for sending your work entitled “Functional topography of the human entorhinal cortex” for consideration at *eLife*. Your article has been favorably evaluated by Eve Marder (Senior editor) and three reviewers, one of whom is a member of our Board of Reviewing Editors.

The Reviewing editor and the other reviewers discussed their comments before we reached this decision, and the Reviewing editor has assembled the following comments to help you prepare a revised submission.

In this well written and interesting study, the authors used 7 Tesla ultra-high field functional magnetic resonance imaging to identify functional subdivisions of the human EC through the analysis of preferred connectivity with parahippocampal and perirhinal cortices. Aside from a solidly supported hypothesis on how connectional patterns might allow the definition of functionally different domains in the human entorhinal cortex, the authors addressed intrinsic connectivity of the EC. This study, as well as its counterpart study by [36], should be welcomed and heralded as an important step forward to our understanding of how to functionally divide EC in humans.

1) Like its counterpart paper, the study falls into a well-known trap of comparative anatomical studies in that chosen nomenclatures are taken to have implications that have never been intended, neither implicit nor explicit. How to define an area in the brain is one problem, how to find the homologue of an area defined in one species in a second species is another question. The authors take an originally cytoarchitectonically defined subdivision of EC into LEC and MEC as the starting point, adding connectivity patterns as the defining second criterion. They then look for connection patterns in the human and find two regions as well. That supports the conclusion that the functional homologues of LEC and MEC can be defined in humans based on the data available (Discussion). In the same sentence it is clearly stated that the LEC homologue is situated anterolateral, while the MEC homologue is to be found at a posteromedial position. Unfortunately, then the trap snaps; the authors then claim that this position is different from that in the rodent since LEC means lateral and MEC means medial. Although they are correct that the given names indicate a position, this is a rather unfortunate interpretation/nomenclature, because in the rodent and in several other species MEC has a posteromedial preferred position and LEC an anterolateral. Maybe a bit less extreme as in the monkey and human, but not all that different. Several papers on the primate brain, including both monkey and human data, unfortunately make the same mistake and assume that all that has a lateral position in EC is LEC and all with a medial position is MEC. The authors correctly conclude that this is likely incorrect, but do not make this explicit and therefore add to the confusion.

A few examples of sentences that add to the confusion instead of clarifying it:

In the Introduction: “Furthermore, the two subregions differ not only in structure and function, but also in their vulnerability to Alzheimer's disease-related pathology (Khan et al., 2013)”. In Khan et al. 2013, the authors indeed use LEC to describe AD pathology, but they actually describe pathology in a longitudinal strip of EC that likely encompasses both LEC and MEC, both when one looks at cytoarchitectonics as well as monkey connectivity studies.

In the Results section: “Based on studies in rodents, the two networks are hypothesised to connect to either the LEC or the MEC, respectively. Studies in non-human primates, however, predict that the two systems project to the anterior and posterior EC, respectively. In order to test both of these predictions…” Here the authors implicitly assume that LEC and MEC are two entities different from alEC and pmEC.

In the Discussion: “Here, we show that a simple correspondence of MEC/LEC to the mediolateral axis in humans is not valid.” This sentence, although literally true, implicitly contradicts the authors' own concluding statement that the functional homologue can be determined based on the outcome.

Also in the Discussion: “Previous neuroimaging studies in humans reported differences between medial and lateral entorhinal cortex that mimicked differences between the rodent MEC and LEC. In light of our findings, these reports could be explained by a partial overlap of the medial and lateral divisions with the pmEC and the alEC, respectively.” Again, medial and lateral are here used as pure topological indicators, not as functional, hodological or cytoarchitural definition, so using LEC and MEC is confusing.

Therefore, we strongly suggest that the authors start off with defining areas as alEC and pmEC, and state that other human studies used LEC and MEC to indicate lateral and medial strips of EC. In the remainder of the manuscript they should only use LEC and MEC when is clear that these are defined, not just indicating a position. This would lead and support their statement in the Discussion: “However, the anatomical organisation and connectivity patterns are largely conserved across species and the characteristic projections from the PRC and PHC rather map on the anteroposterior axis in primates.”

2) The paper can be improved by adding a few simple analyses that can hopefully illustrate the results in a clear manner. Below are potential technical issues that the authors should address. These issues, described in more detail below, should be straightforward to deal with in the revision:

A) The study elegantly demonstrates that pmEC and alEC show different patterns of whole-brain connectivity with extended posterior medial (PM) and anterior temporal (AT) neocortical networks. If we understand the methods, the authors used a single ROI for the entire PM network and one for the entire AT network. At least for illustrative purposes, it would be useful to present separate results for each sphere and each EC ROI. Ideally they could show a correlation matrix with the 4 EC ROIs (well, actually 8, as described in my next point) in columns and the individual PM and then AT ROI spheres in rows. Hopefully, on a quick scan of the matrix, readers will be able to see differences in the connectivity profiles of the anterior lateral and posterior medial EC ROIs.

B) I did not see any mention of laterality. I might have missed it, but if the authors collapsed across hemispheres, I'd like them to consider also presenting results with separate left and right hemisphere ROIs. Doing so would provide a natural internal replication, because one would expect more extensive contralateral connectivity within the alEC regions and within the pmEC regions than between alEC and pmEC.

C) Along the same lines as the first 2 points, it appears that the first set of a priori ROI analyses separately examine medial and lateral EC ROIs and anterior and posterior EC ROIs. If the authors planned to investigate both axes, wouldn't it make sense to include all 4 ROIs in a single factorial analysis with anterior-posterior and medial-lateral factors? If the analysis revealed an interaction, it would strengthen the case that the functional organization of the EC cuts across both axes.

3) It might also help to present whole-brain maps (preferably on a surface rendering) for the contrast between anterior-lateral and posterior-medial ROI connectivity, so that readers can see which regions show differential connectivity. Alternatively, it might be preferable to show single-region connectivity maps.

4A) In general, motion artifact is always a concern with functional connectivity analyses (summarized in [42]), and more so when one considers that the EC is typically a site with significant signal dropout. It is surprising that these issues are not raised in the paper. We would like to know whether the pmEC and alEC (either defined arbitrarily or using the ConGrads approach) have different spatial or temporal SNR values.

B) Also, the authors used ICA to deal with residual motion artifact, but it is not clear that ICA will adequately remove the effects of transient motion spikes while preserving the rest of the timeseries. It would be reassuring if the authors can replicate the results of at least one of the functional connectivity analyses by using motion scrubbing/censoring instead of ICA correction (see [42]).

5A) The ConGrads analysis is not clear. Giving the authors the benefit of the doubt (more so if they can present the simple analyses suggested above), but a better and simpler explanation of this method is needed. It sounds like a promising approach, but given that the method is not in widespread use, readers may be put off by the brief explanation.

B) As a related issue, the meaning of the two “connectopies” was difficult to figure out. It sounds like what the authors are saying is that there are two connectivity gradients, one anterior-posterior, and one that is lateral-medial. If so, it would be nice for the authors to spell this out, and to simplify the corresponding caption text for Figure 2 and Figure 3. Also, if the algorithm is pulling out two connectivity gradients, could the relationship be more simply and intuitively captured by a single al-to-pm gradient. If it does not distort the actual patterns in the data, the authors should consider somehow merging the two connectopies to create simple alEC and pmEC ROIs. At present, it is difficult for the reader to look at the figures and see a figure that convincingly depicts the functional topography that is described in the text.

[Editors' note: further revisions were requested prior to acceptance, as described below.]

Thank you for resubmitting your work entitled “Functional topography of the human entorhinal cortex” for further consideration at *eLife*. Your revised article has been favorably evaluated by Eve Marder (Senior editor), a Reviewing editor, and one of the original reviewers. The manuscript has been improved but there are some remaining issues that need to be addressed before acceptance, as outlined below.

In the revised manuscript, the authors have carefully addressed the comments about potential confusion in nomenclature when comparing the human brain to the non-human primate and rodent brain, in particular with respect to the subdivisions of EC. The manuscript has improved significantly, however there are still two remaining issues.

First, the authors suggested to change their initially chosen nomenclature from anterolateral to posteromedial EC into temporal to septal EC. Though understandable, the choice is very counterintuitive, since the septotemporal nomenclature is typically used for the non-primate hippocampus where one tip of the hippocampus is topologically associated with the septal complex. This is not the case in the primate, and therefore, most authors elected to use an anterior-posterior nomenclature. I strongly urge the authors to maintain this convention, irrespective of their valid argument that the AP axis does not completely reflect the tilted orientation of the hippocampal and parahippocampal regions in the human. From what is said in the Results, it is clear that the authors seem ambivalent about their newly introduced nomenclature.

In addition, in view of the fact that this study nicely complements another study on subdivisions of the human EC by [36], this provides a unique and important opportunity to establish a consistent human terminology for the entorhinal cortex. To really imprint a new terminology, this is the likely one and only chance to do so; it would be great if the two papers concur on a common terminology. Knowing the two author groups are in touch, the implementation of a common terminology should be feasible and highly welcome.

---

## [Author Response]

*1) Like its counterpart paper, the study falls into a well-known trap of comparative anatomical studies in that chosen nomenclatures are taken to have implications that have never been intended, neither implicit nor explicit. How to define an area in the brain is one problem, how to find the homologue of an area defined in one species in a second species is another question. The authors take an originally cytoarchitectonically defined subdivision of EC into LEC and MEC as the starting point, adding connectivity patterns as the defining second criterion. They then look for connection patterns in the human and find two regions as well. That supports the conclusion that the functional homologues of LEC and MEC can be defined in humans based on the data available (Discussion). In the same sentence it is clearly stated that the LEC homologue is situated anterolateral, while the MEC homologue is to be found at a posteromedial position. Unfortunately, then the trap snaps; the authors then claim that this position is different from that in the rodent since LEC means lateral and MEC means medial. Although they are correct that the given names indicate a position, this is a rather unfortunate interpretation/nomenclature, because in the rodent and in several other species MEC has a posteromedial preferred position and LEC an anterolateral. Maybe a bit less extreme as in the monkey and human, but not all that different. Several papers on the primate brain, including both monkey and human data, unfortunately make the same mistake and assume that all that has a lateral position in EC is LEC and all with a medial position is MEC. The authors correctly conclude that this is likely incorrect, but do not make this explicit and therefore add to the confusion*.

*[…] Therefore, we strongly suggest that the authors start off with defining areas as alEC and pmEC, and state that other human studies used LEC and MEC to indicate lateral and medial strips of EC. In the remainder of the manuscript they should only use LEC and MEC when is clear that these are defined, not just indicating a position. This would lead and support their statement in the second paragraph of the Discussion: ‘However, the anatomical organisation and connectivity patterns are largely conserved across species and the characteristic projections from the PRC and PHC rather map on the anteroposterior axis in primates*.’

We thank the referees for raising this important issue and pointing out the need for clarification. Indeed, ‘MEC’ and ‘LEC’ could be considered a somewhat unfortunate nomenclature, because a literal interpretation misrepresents the actual anatomical localization of the cytoarchitectonic regions. In response to the reviewers' comment we have made the distinction between cytoarchitectonic regions and anatomical location explicit and altered the manuscript accordingly to avoid confusion.

Partly in response to Comment 5B (see below), we decided to change the naming of the two regions we have identified from ‘pmEC’ and ‘alEC’ to septal EC (sEC) and temporal EC (tEC). We feel that this is a less ambiguous nomenclature for the two regions. Firstly, the mediolateral distinction is negligible, while the septotemporal division is dominant (see new Figure 2—figure supplement 1, below). Secondly, the septotemporal axis can be identified independently of the orientation of a given MRI acquisition and therefore is preferable to ‘anteroposterior axis’.

We therefore modified text as follows:

Introduction: “In rodents—and on the basis of cytoarchitectonics—the EC is typically ([54]; [6], and [27]) subdivided into two major subregions, the medial- and the lateral entorhinal cortex (MEC and LEC, respectively). The MEC receives inputs about spatial information from parahippocampal cortex (PHC) and the LEC receives item-related information from perirhinal cortex (PRC) ([54]; Deshmukh et al., 2011, Knierim et al., 2014, and Ranganath et al., 2012).

Also in the Introduction: “Despite a wealth of data and marked differences in structure and function of the rodent MEC and LEC evidence for their human homologue remains elusive. […] In contrast, neuroimaging studies on memory in healthy participants (Schultz 1012, Reagh 2014) and participants with preclinical AD (Khan 2013) suggest that the rodent MEC and LEC map on medial and lateral strips of EC in humans.”

Discussion: “While the shape and location of the EC differs between rodents and primates (Witter et al., 1989), the anatomical organisation and connectivity patterns are largely conserved across species (6). However, translational studies on EC subregions faced the problem of identifying homologous regions across species.”

Still in the Discussion section: “Previous neuroimaging studies in humans reported differences between medial and lateral aspects of entorhinal cortex […] Likewise, anterior slices through the temporal EC contain mostly the homologue of the rodent LEC.”

*2) The paper can be improved by adding a few simple analyses that can hopefully illustrate the results in a clear manner. Below are potential technical issues that the authors should address. These issues, described in more detail below, should be straightforward to deal with in the revision*:

*A) The study elegantly demonstrates that pmEC and alEC show different patterns of whole-brain connectivity with extended posterior medial (PM) and anterior temporal (AT) neocortical networks. If we understand the methods, the authors used a single ROI for the entire PM network and one for the entire AT network. At least for illustrative purposes, it would be useful to present separate results for each sphere and each EC ROI. Ideally they could show a correlation matrix with the 4 EC ROIs (well, actually 8, as described in my next point) in columns and the individual PM and then AT ROI spheres in rows. Hopefully, on a quick scan of the matrix, readers will be able to see differences in the connectivity profiles of the anterior lateral and posterior medial EC ROIs*.

We thank the reviewers for this suggestion. We think that it is indeed highly relevant to characterise the connectivity profiles of the suggested human homologue of the rodent MEC and LEC. Initially, we showed differential whole-volume connectivity maps (Figure 2), to meet this goal. However, comment 2A mainly refers to the model-based analysis that required manual segmentation of the EC. We indeed performed two separate analyses. Each time the EC was divided in two halves: Once along the anteroposterior (septotemporal) axis, once along the mediolateral/proximodistal axis. This approach was pursued to ensure comparable number of voxels per parcel and therefore comparable signal-to-noise ratio (SNR) properties per parcel. We have now described the analysis in more detail in the revised text (Results): we created spherical regions-of-interest (ROIs) *with 4 mm radius* around coordinates pertaining to either of the networks (Ranganath et al., 2012, and [34]) (see Supplementary file 1), as well as ROIs for both the medial and lateral half, and anterior and posterior half of the EC to ensure comparable number of voxels per parcel and therefore comparable signal-to-noise ratio (SNR) properties.

Analysing connectivity of 8 manually segmented EC subregions with single spheres on peak coordinates of the two neocortical networks (AT & PM system, reported by [34] is an interesting suggestion but also comes with challenges, both on a biological and analytical level: This approach relies on piece-wise constant connectivity patterns, where both the EC subregions and the neocortical regions have an all-or-none connectivity preference. However, the connectivity profile—as determined by invasive tracing studies—of the rodent MEC and LEC is not fully segregated, but rather shows overlap. For example, all EC regions have reciprocal projections to the frontal lobes (27) and our own data (Figure 2) suggest this is the case in humans as well. [34] used a Gaussian smoothing kernel of 5 mm FWHM for their functional connectivity analyses (in contrast to 2.5 mm FWHM in our study), which might smooth-out some of the fine-grained differences. The power of the model-based approach results from combining time-courses of a large number of voxels to increase effective SNR. The analysis of connectivity between time-courses of single spheres with 8 EC subregions would suffer from significantly decreased effective SNR.

Nevertheless, we performed the suggested analysis (Figure 6). Note that our initial selection of the ∼100 coordinates reported in Table 1 of [34] comprised many unilateral regions. To avoid proximity biases on connectivity with left or right EC regions, we created a new sub-selection of AT & PM regions with bilateral coordinates (4 bilateral spheres per network, i.e. 16 ROIs) and computed connectivity with 8 EC ROIs (4 quadrants in each hemisphere, as suggested in the following point). Figure 6 shows the resulting connectivity matrix. Note that the division lines of the manually created EC quadrants are roughly parallel to the cardinal axes. This is in contrast to the curved and rather diagonal division line identified with the ConGrads method. Therefore, the anterior-medial and posterior-lateral quadrant likely encompass about equal parts of tEC and sEC. Black boxes in Figure 6 outline connectivity of the anterior-lateral and posterior-medial EC quadrants to the AT and PM network. Only these fields can be expected to have a unique, albeit fractional, contribution of the tEC and sEC regions defined in the main study with the data-driven ConGrads approach. Restricting the analysis to these quadrants indeed revealed an interaction between EC subregion and cortical network (repeated-measures ANOVA: network * EC region: F_(1,21)_=6.9 p=0.0155 ). However, this is not readily apparent from Figure R1 itself. Based on the arguments outlined above, we would prefer to not include this analysis in the manuscript.

Author response image 1.Detailed model-based connectivity analyses. Single spheres to EC quadrants across hemispheres.Functional connectivity (partial correlation coefficients) of spherical ROIs of 4 mm radius, centred on a bilateral selection of coordinates reported by [34] to 4 EC subregions in both hemispheres. The non-overlapping EC subregions were segmented manually. X-axis labels: the first letter indicates laterality (l vs r), the second anterioposterior location (a vs p), the third mediolateral location (m vs l). Y-axis labels: AT1 = Anterior inferior temporal gyrus; AT2 = Perirhinal cortex; AT3 = Dorsolateral prefrontal cortex; AT4 = Orbitofrontal cortex; PM1 = Medial posterior occipital cortex; PM2 = Retrosplenial cortex; PM3 = Occipital pole; PM4 = parahippocampal cortex. Last letter indicates laterality (l vs r).**DOI:**
http://dx.doi.org/10.7554/eLife.06738.018

*B) I did not see any mention of laterality. I might have missed it, but if the authors collapsed across hemispheres, I'd like them to consider also presenting results with separate left and right hemisphere ROIs. Doing so would provide a natural internal replication, because one would expect more extensive contralateral connectivity within the alEC regions and within the pmEC regions than between alEC and pmEC*.

We are grateful for the referees' suggestion to test interhemispheric connectivity of the tEC and sEC. As predicted, the tEC is more connected to itself across hemispheres, than to the sEC on either side. This suggests that even though the two different subregions within a hemisphere are in close spatial proximity, they connect much stronger to their homologue region on the contralateral side. We feel that this is an important addition to the article and included these new results in the revised manuscript (new Figure 2—figure supplement 4).

*C) Along the same lines as the first 2 points, it appears that the first set of a priori ROI analyses separately examine medial and lateral EC ROIs and anterior and posterior EC ROIs. If the authors planned to investigate both axes, wouldn't it make sense to include all 4 ROIs in a single factorial analysis with anterior-posterior and medial-lateral factors? If the analysis revealed an interaction, it would strengthen the case that the functional organization of the EC cuts across both axes*.

We thank the reviewers for this suggestion. We used the connectivity matrix created in response to comment 2A (Figure 6) for a full factorial analysis including factors for each axis (mediolateral and anteroposterior). This analysis did not reveal interactions between cortical network and the two axes (repeated-measures ANOVA; network* anteroposterior: F_(1,21)_=1.3, p=0.259; network* mediolateral: F_(1,21)_=1.77, p=0.198; network* anteroposterior * mediolateral: F_(1,21)_=3.4, p=0.079). Based on their anatomical location in rodents, the anterior-lateral and posterior-medial quadrants are most likely to distinguish the LEC and the MEC, respectively. Therefore they should exhibit the strongest difference in connectivity between the AT and the PM network. Focusing the analysis to these quadrants indeed revealed an interaction between EC subregion and cortical network (repeated-measures ANOVA: network * EC region: F_(1,21)_=6.9 p=0.0155 ). Post-hoc t-tests revealed that the anterior-lateral quadrant connected stronger to the AT compared to the PM network (T_(21)_=4.4, p= 0.0003), while the posterior-medial quadrant did not show this pattern (T_(21)_=1.5: p=0.15). As outlined above, we feel that this information is not essential to the manuscript and would prefer to not include it in the revised manuscript.

*3) It might also help to present whole-brain maps (preferably on a surface rendering) for the contrast between anterior-lateral and posterior-medial ROI connectivity, so that readers can see which regions show differential connectivity. Alternatively, it might be preferable to show single-region connectivity maps*.

We thank the reviewers for this suggestion. We agree that whole brain connectivity maps are valuable for illustrating the connectivity profiles of the tEC and sEC. We show whole-volume connectivity maps in Figure 2 and Figure 3. Crucially, these show strong resemblance of the whole brain connectivity maps of PRC and PHC seeds shown by [34] and therefore complement the initial model-based analysis.

*4A) In general, motion artifact is always a concern with functional connectivity analyses (summarized in*
[42]*), and more so when one considers that the EC is typically a site with significant signal dropout. It is surprising that these issues are not raised in the paper. We would like to know whether the pmEC and alEC (either defined arbitrarily or using the ConGrads approach) have different spatial or temporal SNR values*.

Following the advice by the referees, we performed additional analyses on spatial and temporal SNR and described the results in Figure 2—figure supplement 3, and in the Results section:

“…we examined both spatial and temporal signal-to-noise ratio (SNR) of the tEC and the sEC (Figure 2—figure supplement 3). Temporal SNR did not differ between tEC and sEC (T_(21)_=0.2, p=0.83) but spatial SNR did (T_(21)_= 9.7, p<0.001). This was associated with higher signal in the sEC compared to the tEC (mean signal: tEC=4.97; mean signal sEC=7.76; T_(21)_= 38, p<0.001) in the absence of differences in spatial standard deviation (T_(21)_= 1.5, p=0.144). Note, that mean signal was subtracted from time-series prior to all connectivity analyses (see Methods), which makes it unlikely that signal intensity differences affected the connectivity results.”

Furthermore, the ConGrads method converted the voxel-wise signal to percent-signal-change prior to the functional connectivity analyses. Rather than being sensitive to the magnitude of the signal at a given point in time, our analyses rely on the signal variance over time. Therefore we feel that our results cannot be explained by differences in signal magnitude.

*B) Also, the authors used ICA to deal with residual motion artifact, but it is not clear that ICA will adequately remove the effects of transient motion spikes while preserving the rest of the timeseries. It would be reassuring if the authors can replicate the results of at least one of the functional connectivity analyses by using motion scrubbing/censoring instead of ICA correction (see*
[42]*)*.

We thank the reviewers for pointing out the importance of correcting for motion artefacts. Recently it has been shown that ICA-based methods for correcting motion artefacts (such as the FIX approach employed by us) outperform motion scrubbing and spike-regression methods (Pruim et al., 2015, ICA-AROMA: A robust ICA-based strategy for removing motion artefact from fMRI data; Pruim et al., 2015, Evaluation of ICA-AROMA and alternative strategies for motion artifact removal in resting-state fMRI).

We revised our manuscript accordingly and added the following sentence to the Methods section (in the subsection headed “Data pre-processing”):

“Correction for residual motion artefacts was performed with an ICA-based method, which has been shown to outperform motion scrubbing and spike-regression methods both in terms of reproducibility of resting-state networks and conservation of temporal degrees of freedom”.

We also followed the suggestion of the reviewers and applied motion scrubbing in a separate analysis. The ConGrads analysis yields practically the same results with and without scrubbing. In the initial submission, we already excluded 4 out of 26 participants, based on large numbers of framewise displacements (FD) above 0.5 mm. Now, we extended this approach according to the following description in the Methods section:

“Motion scrubbing. First, we determined time points where instantaneous movement ([41], and [42]) exceeded a threshold of 0.5 mm. Then we excluded these time points (volumes) including the one preceding and the two thereafter from subsequent analyses. Hence, per instantaneous movement above threshold, four volumes were removed. On average this resulted in 121 volumes being removed from participant's time-series (range: 4-300).”

In addition, we now describe the outcome in the Results section in the following way:

“We used an ICA-based method for data cleaning (see Methods) that has been shown to efficiently remove residual effects of head motion. However, we additionally repeated the data-driven connectivity analysis after excluding time periods with large head movements (motion scrubbing ([41], and [42]). Motion scrubbing had only minimal effects on the results. Pearson correlation coefficients of the pre- and post scrubbing results were close to 1 and highly significant (gradient one: left R = 0.9964, right R = 0.9958; gradient two: left = 0.9348, right = 0.8816; all p values < 0.001).”

*5A) The ConGrads analysis is not clear. Giving the authors the benefit of the doubt (more so if they can present the simple analyses suggested above), but a better and simpler explanation of this method is needed. It sounds like a promising approach, but given that the method is not in widespread use, readers may be put off by the brief explanation*.

We expanded and simplified the description of the ConGrads analysis (now mostly referred to as data-driven connectivity analysis) in the following way. The Results section now reads as follows:

Therefore, we adopted a complementary approach to trace the dominant modes of functional connectivity change within the EC in a fully data-driven manner ([22]; see Methods). […] The first and second-dominant modes of functional connectivity change were highly reproducible across resting-state sessions (Pearson's R = 0.99, p < 0.001 and Pearson's R = 0.98, p < 0.001, for the dominant and second-dominant modes, respectively).

In addition, we adapted the Methods section in the following way (subsection headed “Data-driven connectivity analysis”):

“To overcome limitations of the seed-based connectivity analysis, we employed ConGrads (22), which allows for tracing the dominant modes of functional connectivity change within a pre-specified region of the brain in a fully data-driven manner. […] For the ROI analyses on the data of the spatial and non-spatial stimulation experiment, the clusters were warped into MNI space.

*B) As a related issue, the meaning of the two “connectopies” was difficult to figure out. It sounds like what the authors are saying is that there are two connectivity gradients, one anterior-posterior, and one that is lateral-medial. If so, it would be nice for the authors to spell this out, and to simplify the corresponding caption text for*
Figure 2
*and*
Figure 3*. Also, if the algorithm is pulling out two connectivity gradients, could the relationship be more simply and intuitively captured by a single al-to-pm gradient. If it does not distort the actual patterns in the data, the authors should consider somehow merging the two connectopies to create simple alEC and pmEC ROIs. At present, it is difficult for the reader to look at the figures and see a figure that convincingly depicts the functional topography that is described in the text*.

We thank the reviewers for pointing this out. We replaced the term ‘connectopy’ with ‘dominant modes of functional connectivity change’.

Unfortunately, we cannot simply merge the two modes of functional connectivity change. As described above, these represent vectors of multiple, spatially overlapping maps that are sorted according to how well they preserve the similarities among the original, high-dimensional connectivity fingerprints. Thus, the first vector represents the dominant mode of connectivity change in the EC, the second represents the second-dominant mode, and so forth. This is exemplified in Figure 4.

In order to improve the written description of the functional topography and the correspondence with the Figures, we created Figure 2—figure supplement 1 and Figure 3—figure supplement 1 that shows coronal slices of the clusters from the dominant mode of connectivity change. In our view, this figure shows that ‘septal EC’ and ‘temporal EC’ is a more accurate naming than ‘pmEC’ and ‘alEC’.

In the Results section, we modified the text as follows:

“Furthermore, in order to identify the potential human homologues of the rodent LEC and MEC, […] but showed an approximately orthogonal orientation relative to the first (Figure 3 and Figure 3—figure supplement 1).”

Furthermore, we adapted the figure captions (Figure 2 and Figure 3), as requested.

[Editors' note: further revisions were requested prior to acceptance, as described below.]

*In the revised manuscript, the authors have carefully addressed the comments about potential confusion in nomenclature when comparing the human brain to the non-human primate and rodent brain, in particular with respect to the subdivisions of EC. The manuscript has improved significantly, however there are still two remaining issues*.

*First, the authors suggested to change their initially chosen nomenclature from anterolateral to posteromedial EC into temporal to septal EC. Though understandable, the choice is very counterintuitive, since the septotemporal nomenclature is typically used for the non-primate hippocampus where one tip of the hippocampus is topologically associated with the septal complex. This is not the case in the primate, and therefore, most authors elected to use an anterior-posterior nomenclature. I strongly urge the authors to maintain this convention, irrespective of their valid argument that the AP axis does not completely reflect the tilted orientation of the hippocampal and parahippocampal regions in the human. From what is said in the Results, it is clear that the authors seem ambivalent about their newly introduced nomenclature*.

*In addition, in view of the fact that this study nicely complements another study on subdivisions of the human EC by Maass et al., this provides a unique and important opportunity to establish a consistent human terminology for the entorhinal cortex. To really imprint a new terminology, this is the likely one and only chance to do so; it would be great if the two papers concur on a common terminology. Knowing the two author groups are in touch, the implementation of a common terminology should be feasible and highly welcome*.

We agree that a common nomenclature with the companion article by Maass et al. is highly desirable. As suggested by reviewer #3, we consulted with Maass and colleagues to find a common naming for the entorhinal subregions that we identified. We agreed that the anterior-posterior division is dominant, but by no means the only characterising feature. Both of our studies also find evidence for a medial-lateral distinction. To avoid oversimplification, we agreed to refer to the regions as anterior-lateral and posterior-medial EC (alEC and pmEC, respectively) and adapted the manuscript accordingly.

For clarification, we added this sentence to the Results section: “In addition to the dominant anterior-posterior distinction, the posterior cluster was located more medially (and to some extend more dorsally) and the anterior cluster was located more laterally (and to some extend more ventrally). Hereafter, we refer to the clusters as posterior-medial EC (‘pmEC’) and anterior-lateral EC (‘alEC’), respectively, consistent with Maass et al. (Maass, submitted for publication).”

Furthermore, we corrected the use of ‘LEC’ and ‘MEC’ by replacing it with ‘lateral parts of EC’ and ‘medial parts of EC’.